# Research on the Solid–Liquid Composite Casting Process of Incoloy825/P110 Steel Composite Pipe

**DOI:** 10.3390/ma17091976

**Published:** 2024-04-24

**Authors:** Hailian Gui, Xiaotong Hu, Hao Liu, Chen Zhang, Qiang Li, Jianhua Hu, Jianxun Chen, Yujun Gou, Yuanhua Shuang, Pengyue Zhang

**Affiliations:** 1Department of Materials Science and Engineering, Taiyuan University of Sciences and Technology, Taiyuan 030024, China; guihailian2000@163.com (H.G.); s202114210065@stu.tyust.edu.cn (X.H.); s202114110031@stu.tyust.edu.cn (H.L.); b202314110024@stu.tyust.edu.cn (C.Z.); 2005022@tyust.edu.cn (J.H.); tyust202403@163.com (J.C.); tyust123456@163.com (Y.S.); 2Department of Intelligent Manufacturing, Shanxi Vocational University of Engineering Science and Technology, Jinzhong 030619, China; 3Department of Transportation and Logistics, Taiyuan University of Sciences and Technology, Taiyuan 030024, China; tyust1298483299@163.com; 4Shanxi Steel Heli New Material Technology Company Limited, Taiyuan 030021, China; pengyue202403@163.com

**Keywords:** solid–liquid composite, composite tube blanks, interface, bond strength, temperature

## Abstract

Bimetallic composites have a wide range of application prospects in various industries. Different bonding temperatures, as one of the influencing factors, directly affect the bonding effectiveness as well as the performance and application of the materials. Using metallurgical bonding techniques ensures a strong bond at the interface of bimetallic materials, resulting in high-quality composite pipe billets. This paper describes an Incoloy825/P110 steel bimetal composite material made by the solid–liquid composite method. By utilizing ProCAST 14.5 software for simulation and deriving theoretical formulas, an initial range of temperatures for bimetallic preparation has been tentatively determined. And this temperature range will be utilized for on-site experiments to prepare bimetallic samples. After the preparation process is completed, samples will be selected. The influence of the external mold temperature on the interface bonding of Incoloy825/P110 steel solid–liquid composite material is studied using an ultra-depth three-dimensional morphology microscope and a scanning electron microscope. Through research, the optimal preheating temperature range for the solid–liquid composite outer mold of Incoloy825/P110 bimetallic composite material has been determined. The casting temperature of the inner mold has a significant impact on the interface bonding of this bimetal composite material. As the casting temperature of the inner mold increases, the interface thickness gradually increases. At lower temperatures, the interface thickness is lower and the bonding is poorer. At higher temperatures, melting may occur, leading to coarse grains at the interface. When the temperatures of the inner and outer molds are within a certain range, a new phase appears at the interface. Indeed, it increases the strength of the interface bonding. Due to co-melting of the bimetal near the interface, element migration occurs, resulting in increased Ni and Cr content at the interface and enhanced corrosion resistance.

## 1. Introduction

Composite materials are formed by combining two or more materials with different chemical or physical properties to create a new material with distinct physical or chemical characteristics. With the advancement of modern technology, composite materials have been widely applied in various important and innovative industries both domestically and internationally, such as bio-composite materials [1], aerospace, maritime transportation, mechanical structural materials [2], and civil engineering [3,4,5]. Recent studies have shown that composite materials are widely used in automotive, underwater, and structural applications. In industrial applications, higher demands are being placed on the comprehensive performance of metal materials, and bimetallic composite materials have the advantages of two different materials [6,7]. Each metal can maintain its independent performance while also improving overall strength, toughness, and corrosion resistance as a whole. Bimetallic composite materials can effectively cope with high-corrosion environments and high-pressure environments caused by factors such as deep-sea conditions, thus gaining wide attention and being widely applied in aerospace, petrochemical, automotive, shipbuilding, and other fields [8].

Composite materials can be produced through methods such as hot rolling, rotary forging for achieving mechanical composites, explosion welding, and overlay welding for metallurgical composites. Centrifugal casting adopts solid–liquid composite technology, which offers advantages such as high efficiency, low cost, high density of castings, and reduced defects such as porosity and slag inclusion. Moreover, due to the metallurgical composite process at the interface, the bonding strength is superior to mechanical composites [9]. There has been a significant amount of research conducted internationally on composite materials. Hu et al. [10] investigated the influence of preparation temperature on the fiber–matrix interface reaction and concluded that composite materials without significant defects can be prepared at 1150 °C. Wen et al. [11] studied the effect of different solid–liquid casting temperatures on nickel-plated TC4/AZ91D bimetal, and found that this material can be successfully prepared within the casting temperature range of 690 °C to 750 °C. As the casting temperature increases, the brittleness of the interface also increases. Li et al. [12] used solid–liquid composite casting technology to prepare copper–aluminum composite materials and found that copper–aluminum composites can be well sintered at 700 °C, with a certain width of transition zone and element interdiffusion layer, achieving metallurgical bonding. Zhang et al. [13] discovered that corrosion is prone to occur near the interface during aluminum alloy/steel composites, indicating that element migration near the interface can lead to changes in performance. Kah P et al. [14] found that the formation of intermetallic compounds has a certain impact on the thermal properties of the composite interface. It can to some extent avoid the formation of defects. Wang et al. [15] conducted research on the neutral layer of the interface during secondary processing of bimetal composite plates, and found that the secondary processing parameters have a certain influence on interface morphology. Meanwhile, numerical solutions were also obtained using boundary element [16,17] and finite element methods [18,19].

Many researchers have conducted extensive studies on composite materials involving lightweight metals, but there has been relatively less research on composite materials combining nickel-based alloys with carbon steel. Using a single metal such as alloy 825 for piping can be costly. By preparing composite materials with 20 steel, not only can costs be reduced, but also the distinctive properties of both metals can be fully utilized. This can simultaneously meet the requirements for corrosion resistance and high strength. To this end, this study used solid–liquid composite technology to prepare composite pipe billets and conducted theoretical analysis on the conduction of temperature with solid–liquid bonding of the composite interface and on the influence on phase transformation, establishing a convective heat transfer model. Secondly, based on the solidus and liquidus values of the outer mold, numerical simulation was used to determine the optimal heating and casting temperature of the outer mold, and optimal process parameters were given. Finally, field experiments were conducted using the obtained process parameters, and microscopic morphology studies were performed on the interface of the produced composite pipe billets, demonstrating that the research findings of this paper can effectively guide practical production.

## 2. Establishing an Interface Temperature Model

The heat conduction differential equation is the basis for numerical simulation calculation of the temperature field during the casting solidification process [20]. It is derived from the law of energy conservation and Fourier’s law of heat conduction [21] and establishes a heat conduction model:(1)$$\begin{array}{c} \frac{1}{r}\frac{\partial }{\partial r}\left(Kr\frac{\partial T}{\partial r}\right)+\frac{1}{r^{2}}\frac{\partial }{\partial \varphi }\left(K\frac{\partial T}{\partial \varphi }\right)+\frac{\partial }{\partial z}\left(K\frac{\partial T}{\partial z}\right)+\dot{q}=\rho c\frac{dT}{dt} \end{array}$$

A temperature field is established under this model. In order to address the heat transfer problem from liquid metal casting into the mold, a computational model for the initial conditions, boundary conditions, material thermal properties, etc., under this temperature field is determined as shown in Equation (2):(2)$$\begin{array}{c} T_{0}\left(x,\;y,\;z,\;0\right)=T_{p} \end{array}$$

In the equation, $T_{p}$ represents the pouring temperature of the liquid metal.

Due to the large size of the casting, during the simulation of the casting process, the interface temperature is calculated using the steady-state temperature field calculation method. And it is assumed that the liquid metal begins to solidify as a whole immediately after filling the cavity. Therefore, a steady-state temperature field model can be used to calculate the initial temperature of the internal liquid metal and the external mold. Equation (3) is a one-dimensional thermal conductivity model [22].
(3)$$\begin{array}{c} \frac{\partial T}{\partial \tau }=\alpha \frac{\partial^{2}T}{\partial x^{2}} \end{array}$$

When the liquid metal enters the mold, at the instant when the high-temperature liquid metal contacts the inner wall of the mold, the internal metal has not yet begun to solidify. At this time, the internal boundary temperature of the mold is $T_{1}$, the initial temperature of the liquid metal entering the mold is $T_{p}$, and the initial temperature of the mold is $T_{0}$. By substituting the initial conditions into Equation (3), the distribution of the internal metal temperature can be obtained as follows:(4)$$\begin{array}{c} T_{M}=T_{1}+\left(T_{p}-T_{1}\right)\mathrm{erf}\left(\frac{x}{2\sqrt{\alpha_{M}t}}\right) \end{array}$$

The distribution of the mold temperature is as follows:(5)$$\begin{array}{c} T_{m}=T_{1}+\left(T_{1}-T_{0}\right)\mathrm{erf}\left(\frac{x}{2\sqrt{\alpha_{m}t}}\right) \end{array}$$

The temperature of the interface nodes between the inner and outer metals is as follows:(6)$$\begin{array}{c} T_{1}=\frac{b_{M}T_{p}+b_{m}T_{0}}{b_{M}+b_{m}} \end{array}$$

In the equation, C is the specific heat capacity of the material, J/(kg·K); 𝜌 is the density of the material, kg/m^3^; 𝜆 is the thermal conductivity of the material, W/(m·K); and T is the harmonic period, typically taken as 24 h.

Some researchers proposed a formula for the temperature of the inner and outer molds. By using their formula, the following equation for calculating the maximum interface temperature can be obtained by substituting it into Equation (6):(7)$$\begin{array}{c} T_{1}=\frac{5\rho_{0}R_{1}^{2}\left(\frac{L_{0}}{3}+H_{0}\right)}{\rho_{1}C_{1}{(R_{2}^{2}-R}_{1}^{2})}-\frac{5H_{1}}{C_{1}}+T_{S} \end{array}$$
where $\rho_{0}$, $\rho_{1}$ are the densities of the inner and outer metal layers, kg/m^3^; $R_{1}$, $R_{2}$ are the outer diameters of the inner and outer metal layers, respectively, m; $L_{0}$ is the latent heat of solidification of the inner liquid metal, J/kg; $H_{0}$ is the superheat energy of the liquid metal, J/kg; $H_{1}$ is the heat absorbed by the outer metal layer, J/kg; $T_{S}$ is the solidus temperature of the outer metal layer, K; and $C_{1}$ is the specific heat capacity of the outer mold metal, J/gK.

## 3. Simulation Analysis

The finite element method simulated the billet manufacturing process and studied the effects of different temperatures of the inner and outer molds on the maximum temperature at the interface junction. This paper takes P110 steel as the material for the outer mold and Incoloy825 as the material for the inner mold, simulating the billet manufacturing process at different temperatures. Modeling was performed using UG 10.0 software and then imported into ProCAST 14.5 software for simulation analysis. The highest temperature point at the interface between the inner and outer molds was obtained to determine a suitable temperature range for billet manufacturing [23].

### 3.1. Finite Element Model

The inner mold for solid–liquid billet manufacturing is a cylindrical shape with a diameter of 102 mm and a height of 150 mm. The designed sprue is also a cylindrical shape with a diameter of 40mm and a height of 180 mm. The model is shown in Figure 1a, and both the inner mold and sprue are made of Incoloy825 material. The outer mold is a cylindrical shape with a diameter of 210 mm and a height of 330 mm. The model is shown in Figure 1b, and it is made of P110 steel. The system adopts top centrifugal pouring [24], with a heat transfer coefficient set at 1000 W/(m^2^·K), centrifugal speed v = 1 rpm, the cooling method selected as air cooling, and the pouring time set to 10 s.

The three-dimensional model was imported into MeshCAST for meshing. The meshing diagram of the casting is shown in Figure 2. After the meshing was completed, there were a total of 8662 quadrilateral elements and 50,419 tetrahedral elements [25].

The phase diagram obtained from the calculation is shown in Figure 3. Incoloy825 has a solidus line temperature of 1272 °C and a liquidus line temperature of 1375 °C. P110 steel has a solidus line temperature of 1127 °C and a liquidus line temperature of 1443 °C. Referring to the solidus line and liquidus line in Figure 3, the casting temperature for the inner mold is set to be in the range of 1430 °C to 1450 °C.

The thermophysical parameters of 825 and p110 were calculated using JMatPro 7.0, as shown in Figure 4, and the values were input into Equation (7). By using the formula, the preheating temperature range for the outer mold was determined to be 1100 °C to 1300 °C. Numerical simulations were conducted by selecting outer mold temperatures of 1100 °C, 1150 °C, 1200 °C, 1250 °C, and 1300 °C.

### 3.2. Simulation Results and Analysis

To ensure that metallurgical bonding is achieved and that the interface temperature between the inner and outer metal layers is maintained above the solidus line temperature of the outer mold and below the liquidus line temperature, it should be kept as close as possible to the liquidus line temperature, ideally in the range of 1250 °C to 1443 °C. It is also necessary to increase the time close to the liquidus line appropriately based on the diffusion behavior of metal atoms in order to enhance the bonding strength. However, an excessively long diffusion time can also affect the performance of the composite material. Therefore, this study designs the inner mold temperature range to be 1430 °C to 1450 °C and the outer mold preheating temperature range to be 1100 °C to 1300 °C. The pouring process is illustrated in Figure 5.

From Figure 5, it can be observed that the inner mold section is completely filled. After air cooling, the overall temperature of the workpiece is lower than the solidus temperature of the metal, indicating convergence of the simulation results. Figure 5a–d shows the temperature variations at different cross-sections during the cooling process after the liquid metal enters the mold. The center temperatures of the inner layer are 1364.3 °C, 1275.4 °C, 1075.2 °C, and 976.4 °C, respectively, while the average temperatures at the interface are 1327.7 °C, 1252.4 °C, 1062.5 °C, and 962.8 °C, respectively. Due to a solidus and liquidus temperature range of 1127 °C~1443 °C for the outer mold metal P110, the interface temperature in Figure 5a,b falls within this range, while the interface temperature in Figure 5c,d is below this temperature. From Figure 5, when the interface temperature is above the solidus temperature of the outer mold metal, it can be inferred that during this time, the region of highest temperature in the inner mold moves downward as the temperature decreases. When the interface has solidified, the region of highest temperature in the inner mold moves upward. It can be observed that the region of highest temperature in the inner mold metal exhibits a trend of initially moving downward and then upward. At this point, due to the different positions of the center temperature, the variation of interface temperature differs. To more accurately reflect the temperature variation at the interface, it is advisable to select and analyze the interface at the middle section of the inner mold. As shown in Figure 6, the highest temperature at the interface is simulated and presented in Table 1.

From Table 1, it can be observed that the highest temperature at the interface is positively correlated with the outer mold preheating temperature and the liquid metal temperature of the inner mold. From the perspective of bonding temperature, when the outer mold temperature is 1100 °C, the highest temperature at the interface remains below 1250 °C. Due to factors such as heat transfer and loss during the casting process, lower temperatures and shorter composite times may lead to poor interface bonding performance. Therefore, this preheating range is not conducive to composite formation. When the outer mold temperature is at 1300 °C, the outer mold metal is close to the liquidus line temperature, which can lead to the occurrence of leakage. When the outer mold temperature is at 1250 °C, although it exceeds the solidus line temperature of P110, the temperature difference is relatively small. At this temperature, it is conducive to interface bonding. Therefore, the optimal preheating temperature for the outer mold is between 1150 °C and 1250 °C.

## 4. Experimental Analysis

According to the theoretical calculations and simulations of the ingot temperature, casting composites of Incoloy825 and P110 steel are carried out. The composite experiment uses an electromagnetic induction heating device for the outer mold P110 steel, with a heating temperature of 1200 °C. The inner core Incoloy825 is melted using electric melting, and experiments are conducted at temperatures of 1430 °C, 1440 °C, and 1450 °C. The composite is then cooled in air. A schematic diagram of the experimental process is shown in Figure 7.

To study the microstructure at the interface of the Incoloy825/P110 steel composite, the prepared composite billet is cut and sampled at the billet interface, as shown in Figure 8. Figure 8a shows the composite interface state at 1430 °C of the internal mold, at which point there are noticeable cracks at the interface, indicating poor bonding. Figure 8b shows a well-bonded area, where the generation of new phases can be clearly seen macroscopically. Figure 8c shows the over-melted zone, where the interface is not clearly defined. After cutting, the samples selection near the interface are shown in Figure 8d.

The sample is cut into a size of 10 mm × 10 mm and the surface of the sample is mechanically polished after grinding. Afterwards, a solution of 10 g oxalic acid with 100 mL H_2_O [26] and a solution of nitric acid with alcohol are used to corrode the Incoloy825 side and the P110 steel side, respectively. Subsequently, the samples are washed with 95% alcohol, dried with a blower, and then subjected to metallographic experiments. The elemental composition at the interface is analyzed using an energy dispersive spectrometer (EDS).

Using a high-depth-of-field 3D profilometer, images of the billet interface at different temperatures were captured at a magnification of 200 times, as shown in Figure 9(a_1_–c_1_), are metallographic images at outer mold temperatures of 1200 °C and inner mold temperatures of 1430 °C, 1440 °C, and 1450 °C, respectively. Figure 9(a_2_–c_2_) shows the corresponding side scan images of the P110 steel sample, Figure 9(a_3_–c_3_) shows the corresponding side scan images of the Incoloy825 sample, and Figure 9(a_4_–c_4_) shows line scan count images at the corresponding sample interface. According to the metallographic images, after corrosion, the left side of the interface is P110 steel, mainly characterized by a large amount of ferrite and a small amount of a new phase. The right side is Incoloy825, mainly characterized by austenite [27]. According to the line scan count images at the interface, as shown in Figure 9(a_4_–c_4_), it can be observed that there are changes in the content of representative elements Fe, Ni, and Cr. Specifically, Fe content increases significantly from the Incoloy825 side to the P110 steel side, while Ni and Cr elements decrease noticeably. This indicates the occurrence of element migration at the interface and the formation of new phases. As the temperature increases, the thickness of the bonding interface gradually increases.

Figure 9(a_1_) represents the initial bonding state, and it can be observed that the thickness of the bonding interface is only about 10.7 μm, indicating a relatively small bonding thickness and weak bonding strength at the interface. Figure 9(b_1_) shows a well-bonded area, where the bonding interface thickness is approximately 715 μm, indicating a relatively thick layer of a new phase and a good bonding effect at the interface. Upon closer examination, new phases are found at the interface. It can be observed that the new phases are relatively fine. This indicates that the newly formed phase at the interface exhibits higher strength. In Figure 9(c_1_), the interface exhibits clear signs of overheating, with a bonding thickness of approximately 963.3 μm. A new phase is formed, but on the left side of the interface, austenite grains can be clearly observed, indicating the presence of pure Incoloy825 metal. On the right side of the interface, there is evident ferrite, which could affect the metallic properties of the Incoloy825 side. Moreover, the grain structure at the interface is coarser compared to that in Figure 9b_1_, resulting in lower strength.

In this study, a solid P110 steel outer mold was used for casting Incoloy825 melt. Therefore, the metals near the bimetallic interface experience alloying and exhibit an element migration phenomenon [28].

To compare the diffusion of Cr and Ni elements at the interface, distant areas of Incoloy825 and P110 steel can be considered as single metals without element diffusion. As shown in Figure 10, EDS surface scans were performed on these areas, and the results are shown in Table 2.

According to Table 2 and Figure 9, it can be observed that the content of Ni and Cr in the Incoloy825 metal near the interface is lower compared to the single metal, while the content of Ni and Cr in the P110 metal near the interface is higher compared to the single metal.

For the internal mold temperature at 1440 °C, it can be inferred from a comparison between Figure 9b_2_ and Figure 9b_3_ that there is a clear decrease in Cr and Ni elements in the Incoloy825 at the interface, while there is a noticeable increase in Cr and Ni elements in the P110 steel side. This indicates that there has been diffusion of Cr and Ni elements at the interface, and also suggests that the new phase at the interface contains a higher content of Cr and Ni elements. The elements Cr and Ni can enhance the strength, hardness, and wear resistance of metals, while also improving the corrosion resistance and heat resistance of steel.

It is noteworthy that carbon exhibits diffusion behavior during the formation of bimetallic composites. The research suggests that carbon exhibits a pronounced tendency for diffusion and segregation, while also reacting with other elements. As a result, different carbide phases are formed at the end of solidification [29]. Having performed line scan experiments at well-bonded interface locations, Table 3 shows the elemental composition.

From Table 3, it can be observed that the atomic ratio of Cr and Mo elements to C elements at the interface is close to 7:3. Research indicates that both Cr and Mo atoms readily form carbides with carbon, consistent with the atomic ratio of M_7_C_3_ carbide in the phase diagram. This also suggests the diffusion of elements at the interface.

At the interface, the majority of the new phases consist of the combination of C from P110 and Cr and Ni elements from Incoloy825. Observation of the microstructure at the interface magnified to 500 times is shown in Figure 11. From the figure, it can be observed that there is a significant presence of lamellar and rod-like structures. This structure is mainly composed of M7C3 carbides, which can improve the toughness of the material to some extent.

It can be observed that metallurgical bonding between Incoloy825 and P110 steel has been achieved, and the interface exhibits high toughness, making it less susceptible to cracking. Based on the above analysis, the new phase at the interface exhibits the combined properties of both metals, which can effectively extend the service life of the composite pipe billet.

## 5. Conclusions

This study focuses on controlling the temperature of a solid–liquid composite pipe billet. To achieve better metallurgical composite results, it is necessary to control the interface temperature between the solid phase line and the liquid phase line of the outer mold. The aim is to facilitate adequate atomic diffusion, achieving a higher interface bonding strength. At the same time, excessive atomic diffusion must be prevented. This could compromise the material’s performance and lead to a decrease in the corrosion resistance of the metal on one side, affecting its usability.

The research findings indicate that the optimal preheating temperature for a solid–liquid composite external mold of Incoloy825/P110 steel bimetal composite material is 1150 °C to 1250 °C. The casting temperature of the internal mold has a significant influence on the bonding interface of Incoloy825/P110 bimetal composite material. With an increase in the casting temperature of the internal mold, the interface thickness gradually increases. When the temperature is low, the interface thickness is low, resulting in poor bonding. At higher temperatures, an over-melting phenomenon may occur, and the grains at the interface are coarse. There are implications for the subsequent preparation of composite pipes. At an outer mold preheating temperature of around 1200 °C and an inner mold preheating temperature of around 1440 °C, a significant amount of M_7_C_3_ carbide appears at the interface. This indicates that there is elemental diffusion occurring between the P110 side and the Incoloy825 side, leading to the formation of new phases. The newly formed phases exhibit good toughness, providing a good performance billet for subsequent processes such as piercing rolling. At the same time, due to the co-melting of metals near the interface of Incoloy825 and P110 bimetal, element migration occurs, leading to an increase in Ni and Cr content at the interface of Incoloy825/P110 steel bimetal composite material, enhancing its corrosion resistance.

In this study, only the phase composition at the interface of the middle section of the billet was studied, which may introduce certain errors. In the next step, we will conduct research on different parts of the billet and analyze their influencing factors. We also plan to study the mechanical properties at the interface. This could reflect the performance quality of finished composite pipe products.

## Figures and Tables

**Figure 1 materials-17-01976-f001:**
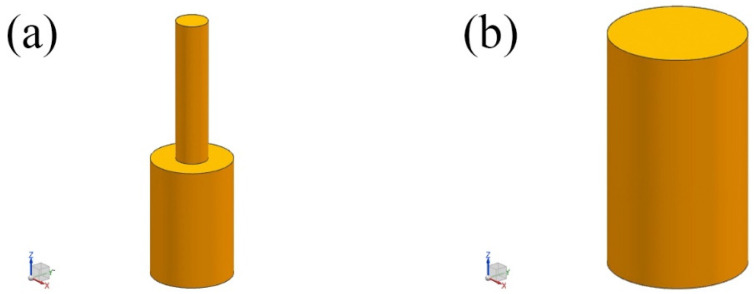
A three-dimensional model of the inner and outer molds. (**a**) A model of the inner mold and gating system; (**b**) a model of the outer mold.

**Figure 2 materials-17-01976-f002:**
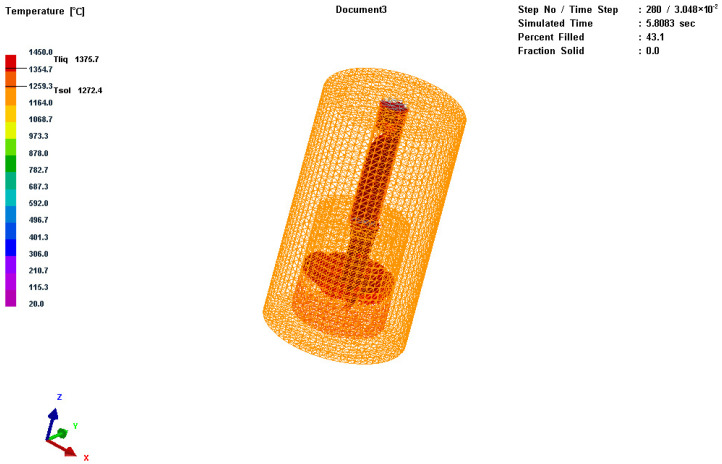
Simulation image of mesh division for the inner and outer molds.

**Figure 3 materials-17-01976-f003:**
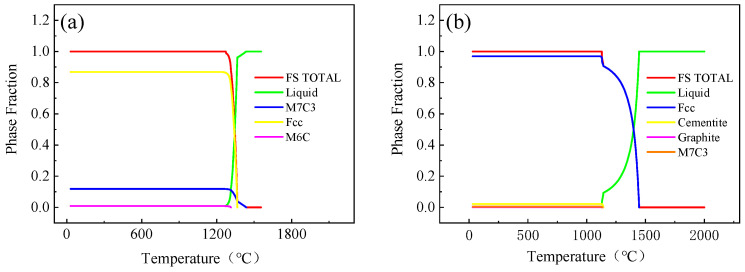
Incoloy825 and P110 steel phase diagrams. (**a**) A phase diagram for Incoloy825; (**b**) a phase diagram for P110 steel.

**Figure 4 materials-17-01976-f004:**
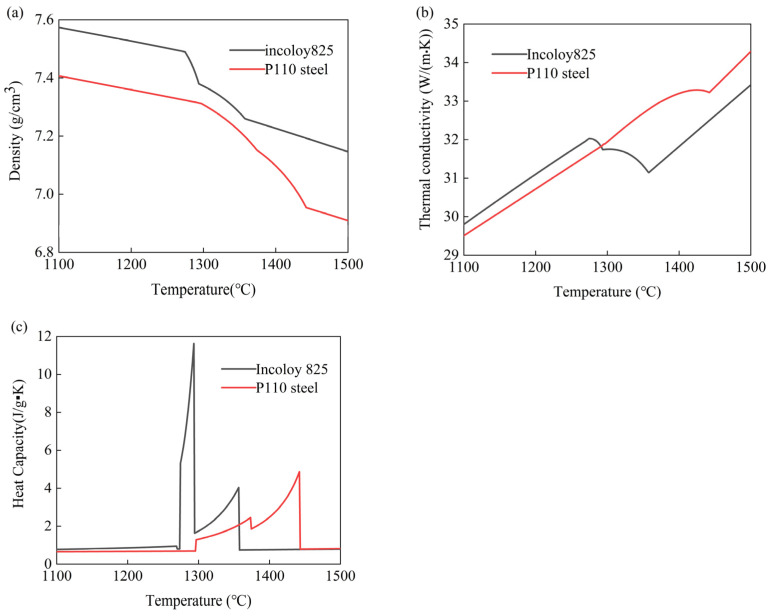
Incoloy825 and P110 steel thermal properties parameters. (**a**) Density, (**b**) thermal conductivity, and (**c**) thermal diffusivity.

**Figure 5 materials-17-01976-f005:**
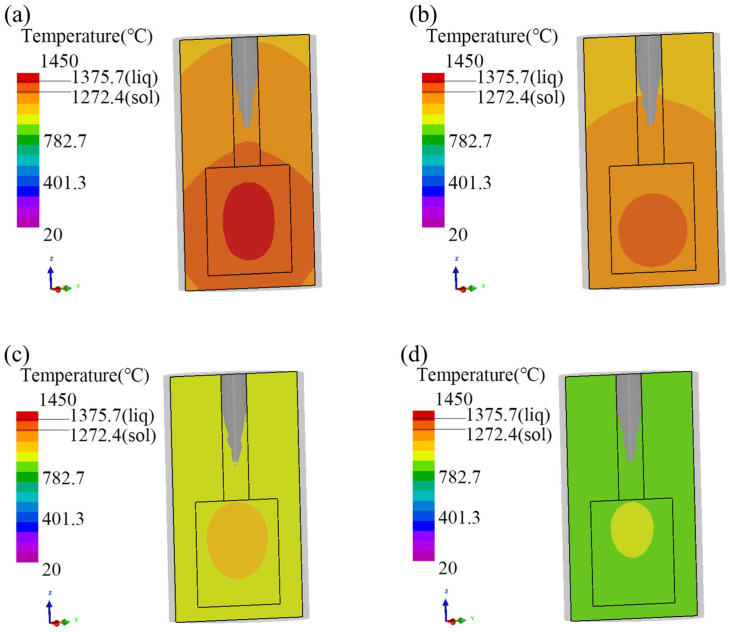
Temperature distribution of the pouring process. (**a**) The core temperature is 1364.3 °C, (**b**) The core temperature is 1275.4 °C, (**c**) The core temperature is 1075.2 °C, (**d**) The core temperature is 976.4 °C.

**Figure 6 materials-17-01976-f006:**
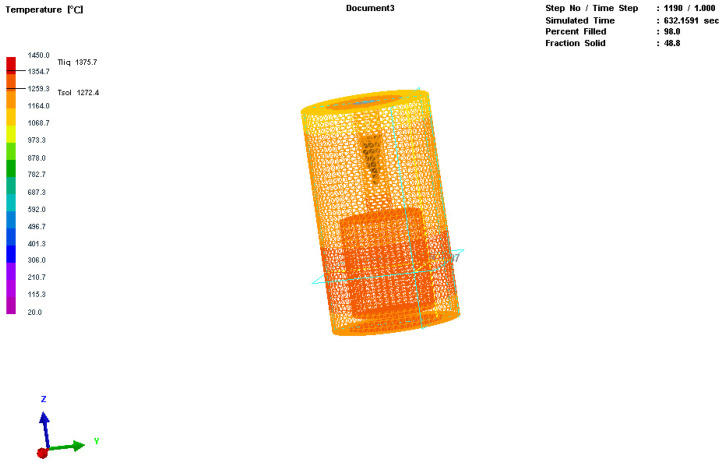
An image of interface point selection.

**Figure 7 materials-17-01976-f007:**
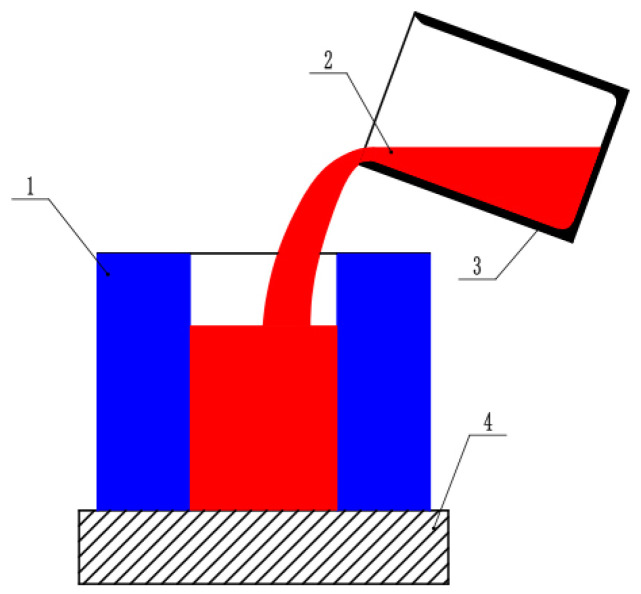
Experimental schematic image. (1) Solid-state P110 steel, (2) liquid Incoloy825, (3) pouring illustration, and (4) detachable bottom support.

**Figure 8 materials-17-01976-f008:**
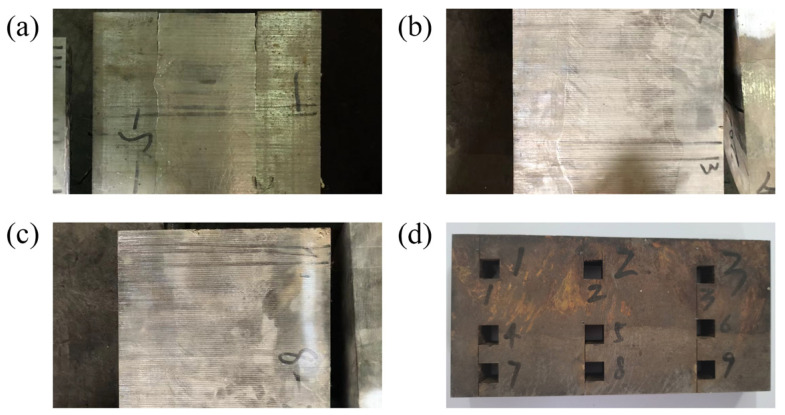
Sample collection. (**a**) The initial bonding state; (**b**) a well-bonded area; (**c**) an over-melted area; (**d**) a section sampling diagram.

**Figure 9 materials-17-01976-f009:**
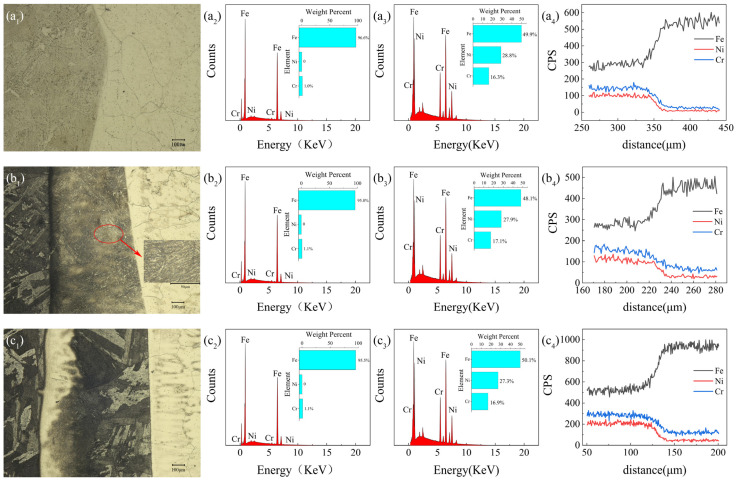
Metallographic image and EDS experimental characterization. (**a_1_**–**c_1_**) are metallographic diagrams of the outer mold temperature of 1200 °C, the inner mold temperature of 1430 °C, 1440 °C and 1450 °C, respectively, (**a_2_**–**c_2_**) are side scans of the corresponding sample P110, (**a_3_**–**c_3_**) are side scans of the corresponding sample Incoloy825, (**a_4_**–**c_4_**) are the line scan count plots at the interface of the corresponding sample. The red circle is at the interface, leading to a larger image.

**Figure 10 materials-17-01976-f010:**
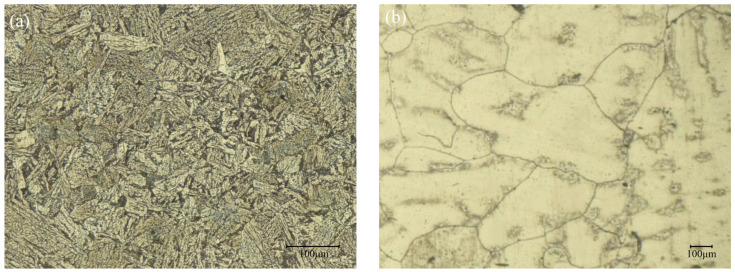
Single metal images. (**a**) Single P110 steel; (**b**) single Incoloy825.

**Figure 11 materials-17-01976-f011:**
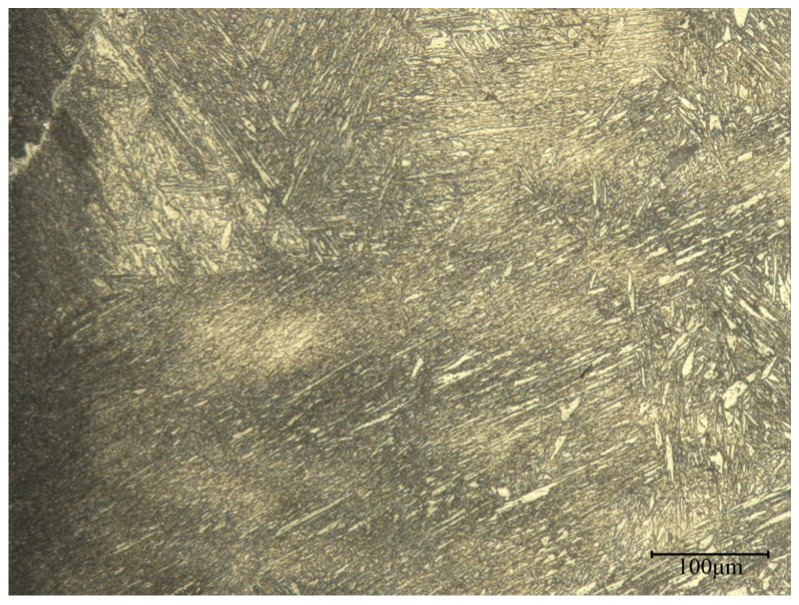
Metallographic diagram at the interface.

**Table 1 materials-17-01976-t001:** Different simulations of inner and outer mold temperatures.

Temperature of P110 Steel (℃)	Maximum Temperature of the Interface at Different Internal Mold Temperatures (℃)		
	1430	1440	1450
1100	1242.41	1244.76	1245.51
1150	1272.67	1275.09	1276.45
1200	1291.15	1292.73	1293.85
1250	1308.4	1310.24	1311.61
1300	1332.46	1334.78	1336.5

**Table 2 materials-17-01976-t002:** Composition of Fe, Cr, and Ni elements in Incoloy825 and P110 steel (%).

Material	Fe	Cr	Mo	Ni	C
825	41.4	18.9	2.5	33.4	1.3
P110	96.4	1.0	0.7	0	1.1

**Table 3 materials-17-01976-t003:** Elemental composition at the interface (%).

Element	Fe	Cr	Mo	Ni	C
content	78.7	1.0	1.2	17.1	0.9

## Data Availability

Data are contained within the article.

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
