# Peer review of "Research on the Solid–Liquid Composite Casting Process of Incoloy825/P110 Steel Composite Pipe"

_materials, 2024, doi:10.3390/ma17091976_

Round 1

Reviewer 1 Report

Comments and Suggestions for Authors

This paper has low novelty and academic depth, especially for the diffusion interface and microstructures of coloy825/P110 steel Composite Pipe. According the simulated phase diagram, it should have many phase, e.g. Fcc, , Liquid, M7C3, M6C, Cementite, and Graphite. Therefore, these phases were not discussed in this paper. Because these phases will diffuse during the composite casting. I think that it is very important to the phase transformation during the composite casting.  Therefore, I gave a major revision for this manuscript.  I suggest that the authors can add some data (metallogray, EDS analysis of phases) about the microstructural observation and discussion of M7C3, M6C, Cementite, and Graphite and explain theris phase transformation of these phases during the composite casting.   

Author Response

Line 314-329 A series of interpretations of M7C3 carbide are performed.

Reviewer 2 Report

Comments and Suggestions for Authors

In the article  "Research on the Solid Liquid Composite Casting Process of Incoloy825/P110 Steel Composite Pipe" the authors  present a detailed study on the preparation of Incoloy825/P110 steel bimetal composite material using a solid-liquid composite method. The authors explore the effects of external mold temperature on the interface bonding of this composite material through several  analytical methods, including ProCAST software simulation, ultra-depth three-dimensional morphology microscope, and scanning electron microscope.

·       The scale bars in Figure 7  are not visible.

·       the novelty of the work should be highlighted better. The authors should provide a more detailed discussion on how this research differs from existing studies in the field of solid-liquid composite casting,

·       The authors should rewrite the conclusion section. It is  important that the authors present the findings avoiding bullet points or lists. The revised section should seamlessly integrate the study's key outcomes.

·       How do the authors interpret the relationship between the mold temperatures and the quality of interface bonding?

·       What are the limitations of the experimental design, and how might they impact the results?

Comments on the Quality of English Language

extensive revision 

Author Response

  • A scale bar has been added
  • Line79-84 The innovation section of this study is written.
  • Line33-366 The conclusion has been revised
  • Line196-215 Simulation analysis is complemented
  • Detail the limitations of the study at the conclusion

Reviewer 3 Report

Comments and Suggestions for Authors

The paper is focused on Incoloy825/P110 steel composite material, and particularly with the method of optimally producing this composite by means of solid-liquid composite technology. It performs both numerical and experimental analysis.

The paper is generally well structured. It would be definitely of interest to the research community in this field of work. However, it demands serious improvements in order to be acceptable for publishing. The following comments must be suitably addressed prior to any final decision:

1) The quality of English language must be significantly improved. The presence of quite basic grammar errors and occasionally a rather weak writing style call for professional assistance, which is strongly suggested. Just a few examples here:

Line 14: “Abstract: This paper prepares Incoloy825/P110 steel…” The paper does not prepare. The papers deals with, describes, etc., but does not prepare.

Line 84: “basic basis”

Line 89: “Establishing the temperature field under this model.” This is not a proper sentence.

Lines 99-100: “According to the one-dimensional heat conduction model as shown in Equation (3).” This is not a proper sentence.

Too many places in the manuscript miss empty spaces.

Etc.

2) Abstract needs to be rewritten. The authors go too much into details about results. No sufficient information about the methods and motivation. The major, general findings should be mentioned, without too many details. This needs to be left for the paper.

3) In the beginning of the abstract, the authors emphasize the significance of composite materials. Some recent works in this field should be mentioned:

Phiri, R., Rangappa, S., Siengchin, S., & Marinkovic, D. (2023). Agro-waste natural fiber sample preparation techniques for bio-composites development: methodological insights. Facta Universitatis-Series Mechanical Engineering, 21(4), 631-656. doi:https://doi.org/10.22190/FUME230905046P

Elmoghazy, Y., Safaei, B., & Sahmani, S. (2023). Finite element analysis for dynamic response of viscoelastic sandwiched structures integrated with aluminum sheets. Facta Universitatis-Series Mechanical Engineering, 21(4), 591-614. doi:https://doi.org/10.22190/FUME231004045E

4) Line 52: The authors should avoid the usage of the term “domestically”, as this is an international journal. I believe, they want to say “in China”.

5) Lines 85-86: “…mainly derived…” What else has been used in derivation?

6) Lines 95-97: “In simulating the casting process, the relative flow of the liquid metal and radiation heat transfer are ignored, and it is assumed that the liquid metal begins to solidify as a whole immediately after filling the cavity.” What justifies neglecting of the mentioned effects and the assumptions made?

7) Line 106: “By substituting the initial and boundary conditions into the equation (3)…” Eq. (3) does not contain boundary conditions per se. This is differential equation and it needs first an ansatz, a general form of solution with some constants that would be resolved using boundary and initial conditions.

8) Quite basic: when writing m3, 3 must be written as a superscript.

9) Line 120 should not be indented, this is not a new paragraph. What is ro11?

10) Line 127: FEM does not do anything “intuitively”. This is a totally wrong interpretation. This is a numerical method that resolves partial differential equations and it is based on physics and mathematics, not on intuition.

11) Add empty spaces between the numbers and the units.

12) Line 150: “… 8662 surface meshes and 50419 volume meshes…” The authors should check the basic FEM terminology. I assume they refer here to finite elements, not to the “meshes”.

13) Have you performed convergence analysis?

14) Lines 156 and 157: Why do the authors use imperative to write the text (calculate, substitute…)?

15) Figure 2 is not a diagram. Later figures represent diagrams, but this one does not. Text in the figure is not readable, which is unacceptable. Also, Figure 7 contains some rather small text.

16) Conclusions must be written better. They should not start with numbers. Also, the numbers are doubled. Conclusions must mention limitations of the work and also directions of the future work.

17) Double numbers in references? Many empty spaces missing in references.

Comments on the Quality of English Language

The comments are provided in the comments for authors. 

Author Response

  • Line 18: “prepares”in “This paper prepares Incoloy825/P110 steel…” changed to“describes”

Line 96: remove “basic”

Line 101: “Establishing the temperature field under this model.” changed to “The temperature field is established under this model.”

Lines112: “According to the one-dimensional heat conduction model as shown in Equation (3).” changed to “Equation (3) is a one-dimensional thermal conductivity model.”

  • Line14-34 Make edits to the summary
  • Line42-43 remove “biomedicine”å’Œ“automotive”,add“Bio-composite Materials”, “Mechanical Structural Materials” and add references
  • Line 59-60: changed to “There has been a significant amount of research conducted internationally on composite materials.”
  • Lines 97-98: “mainly derived from the law of energy conservation and Fourier's law of heat conduction” changed to “It is derived from the law of energy conservation and Fourier's law of heat conduction”
  • Line107-109 “In simulating the casting process, the relative flow of the liquid metal and radiation heat transfer are ignored,” changed to “Due to the large size of the casting, during the simulation of the casting process, the interface temperature is calculated using the steady-state temperature field calculation method”
  • Line 118:remove “and boundary”
  • Lines 126:“m3” changed to “m3

Lines 132:“m3” changed to “m3

  • There is no indentation in the text, and I can't find where ro11 is.
  • Line 139 remove “intuitively”
  • Line 150 Spaces have been added between numbers and units

Line 151 Spaces have been added between numbers and units

Line 153 Spaces have been added between numbers and units

  • Line 162:“surface meshes” changed to “quadrilateral elements”,“volume meshes” changed to “tetrahedral elements”
  • Line 196-215 Simulation analysis and convergence analysis are added.
  • Line 168 “Calculate the thermal parameters of Incoloy825 and P110 steel by using JMatPro” changed to “The thermophysical parameters of 825 and p110 were calculated by using JMatPro”

Line 169 “Substitute the numerical values into Equation (11)”改为“and bring the value into the equation (11)”

  • Figure 2: “diagram” changed to “image”

Figure 7: “diagram” changed to “image”

Figure 9: “diagram” changed to “image”

  • Line338-366 The conclusion has been revised
  • Reference figures have been revised.

Round 2

Reviewer 1 Report

Comments and Suggestions for Authors

I have checked the revised manuscript which was well written after major revision. Therefore, I recommend that this paper can be accepted for publication in Materials without further revision.  

Reviewer 2 Report

Comments and Suggestions for Authors

I recommend the publication of this article.

Comments on the Quality of English Language

moderate revision

Reviewer 3 Report

Comments and Suggestions for Authors

The manuscript has been suitably revised. It is recommended for publishing as it is.